# Triphenylphosphonium Analogs of Short Peptide Related to Bactenecin 7 and Oncocin 112 as Antimicrobial Agents

**DOI:** 10.3390/pharmaceutics16010148

**Published:** 2024-01-22

**Authors:** Andrey G. Tereshchenkov, Zimfira Z. Khairullina, Inna A. Volynkina, Dmitrii A. Lukianov, Pavel A. Nazarov, Julia A. Pavlova, Vadim N. Tashlitsky, Elizaveta A. Razumova, Daria A. Ipatova, Yury V. Timchenko, Dmitry A. Senko, Olga V. Efremenkova, Alena Paleskava, Andrey L. Konevega, Ilya A. Osterman, Igor A. Rodin, Petr V. Sergiev, Olga A. Dontsova, Alexey A. Bogdanov, Natalia V. Sumbatyan

**Affiliations:** 1Department of Chemistry, Lomonosov Moscow State University, 1/3 Leninskie Gory, 119991 Moscow, Russiazkh_msu@mail.ru (Z.Z.K.); inna-volynkina@yandex.ru (I.A.V.); dmitrii.a.lukianov@gmail.com (D.A.L.); elizaveta_razumova@list.ru (E.A.R.); osterman@yandex.ru (I.A.O.); petya@belozersky.msu.ru (P.V.S.); dontsova@belozersky.msu.ru (O.A.D.);; 2A.N. Belozersky Institute of Physico-Chemical Biology, Lomonosov Moscow State University, 1/40 Leninskie Gory, 119991 Moscow, Russia; 3Shemyakin-Ovchinnikov Institute of Bioorganic Chemistry, Russian Academy of Sciences, 117997 Moscow, Russia; 4Gause Institute of New Antibiotics, 11 B. Pirogovskaya Street, 119021 Moscow, Russia; ovefr@yandex.ru; 5Molecular and Radiation Biophysics Division, Petersburg Nuclear Physics Institute, NRC “Kurchatov Institute”, 188300 Gatchina, Russiakonevega_al@pnpi.nrcki.ru (A.L.K.); 6Institute of Biomedical Systems and Biotechnology, Peter the Great St. Petersburg Polytechnic University, 195251 St. Petersburg, Russia; 7NBICS Center, NRC “Kurchatov Institute”, 123182 Moscow, Russia; 8Institute of Functional Genomics, Lomonosov Moscow State University, 119991 Moscow, Russia

**Keywords:** alkyl(triphenyl)phosphonium, proline–arginine-rich antimicrobial peptides, bactenecin 7, oncocin 112, bacterial ribosome, bacterial membrane potential, antimicrobial activity

## Abstract

Antimicrobial peptides (AMPs) have recently attracted attention as promising antibacterial agents capable of acting against resistant bacterial strains. In this work, an approach was applied, consisting of the conjugation of a peptide related to the sequences of bactenecin 7 (Bac7) and oncocin (Onc112) with the alkyl(triphenyl)phosphonium (alkyl-TPP) fragment in order to improve the properties of the AMP and introduce new ones, expand the spectrum of antimicrobial activity, and reduce the inhibitory effect on the eukaryotic translation process. Triphenylphosphonium (TPP) derivatives of a decapeptide RRIRPRPPYL were synthesized. It was comprehensively studied how the modification of the AMP affected the properties of the new compounds. It was shown that while the reduction in the Bac7 length to 10 a.a. residues dramatically decreased the affinity to bacterial ribosomes, the modification of the peptide with alkyl-TPP moieties led to an increase in the affinity. New analogs with structures that combined a decapeptide related to Bac7 and Onc112—Bac(1–10, R/Y)—and TPP attached to the C-terminal amino acid residue via alkylamide linkers, inhibited translation in vitro and were found to be more selective inhibitors of bacterial translation compared with eukaryotic translation than Onc112 and Bac7. The TPP analogs of the decapeptide related to Bac7 and Onc112 suppressed the growth of both Gram-negative bacteria, similar to Onc112 and Bac7, and Gram-positive ones, similar to alkyl-TPP derivatives, and also acted against some resistant laboratory strains. Bac(1–10, R/Y)-C2-TPP, containing a short alkylamide linker between the decapeptide and TPP, was transferred into the *E. coli* cells via the SbmA transporter protein. TPP derivatives of the decapeptide Bac(1–10, R/Y) containing either a decylamide or ethylamide linker caused *B. subtilis* membrane depolarization, similar to alkyl-TPP. The Bac(1–10, R/Y)-C2-TPP analog was proven to be non-toxic for mammalian cells using the MTT test.

## 1. Introduction

AMPs have recently attracted the attention of many researchers as promising antibacterial agents for medical uses [1]. This interest is caused by the ability of AMPs to act on resistant bacteria, the low bacterial resistance against them, the wide range of antimicrobial activities, the variety of mechanisms of action, and relatively simple approaches for modifying their structures [2]. Proline–arginine-rich antimicrobial peptides (PA-AMPs) include peptides with a high content of proline and a PR motif in their primary structure. They have a non-lytic mechanism of action on bacteria and are present in a wide range of organisms, from insects to mammals, being the products of their innate immune systems and providing protection against the penetration of bacteria [3,4,5,6]. The mechanism of the antibacterial action of these peptides is associated with intracellular targets, and as has been shown, PA-AMPs bind to ribosomes and inhibit protein synthesis [7,8,9].

Depending on the interaction with the ribosome and the mechanism of translation inhibition, PA-AMPs are divided into two types. The first group includes oncocins, bactenecins, and some other peptides. These peptides bind inside the nascent peptide exit tunnel (NPET) in the opposite direction to the nascent peptide chain in such a way that the N-terminus is located at the peptidyl transferase center (PTC) and the C-terminus extends deep into the NPET. PA-AMPs in complex with ribosomes allow the formation of the first peptide bond but prevent the elongation of the nascent peptide [10,11,12,13,14]. The second group of studied PA-AMPs includes natural apidaecin peptides and their synthetic analogs, as well as drosocin, which bind to the ribosome in the same orientation as the nascent peptide chain. These AMPs cause ribosome stalling at the stop codon during translation as a result of their interaction with class 1 release factors (RFs) in the PTC, which leads to the depletion of the number of free RFs in the bacterial cell [15,16,17,18]. The mammalian PA-AMP Bac7 identified in cows (*Bos taurus*) and then in sheep (*Ovis aries*) and goats (*Capra hircus*) [6,12] as well as the insect PA-AMP oncocin from the milkweed bug *Oncopeltus fasciatus* [19] are among the most well-studied PA-AMPs [7,20,21].

The main problem with the use of AMPs as potential drugs is their short half-lives in biological media as a result of proteolytic degradation [2]. A number of oncocin analogs have been synthesized, containing various substitutions of amino acid residues at all 19 positions of the peptide chain, as well as double and triple substitutions, to improve their inhibitory activity and proteolytic stability. Among them, Onc112 has turned out to be the most optimal (Figure 1A) [22]. Mammalian PA-AMPs are generally longer than insect PA-AMPs [7]. Shorter analogs of Bac7, whose sequence consists of 60 amino acid residues, have been obtained to identify the domain responsible for its antimicrobial activity. The N-terminal peptides Bac7(1–35) and Bac7(1–16) (Figure 1A) were shown to retain antibacterial activity [23]. The crystal structures of the bacterial ribosome in complex with Onc112 and Bac7(1–16) revealed that despite the different origins of these peptides, they interact with similar regions of the ribosome. Both Bac7 and Onc112 compete effectively with other antibiotics targeting the NPET, such as macrolides, chloramphenicol, lincosamides, and some others [10,11,12,14]. These works revealed in detail the mechanism of action of type 1 PA-AMPs on the bacterial translation process, as well as showing that their relatively short N-terminal sequences are essential for binding to ribosomes.

Despite the structural similarity of Onc112 and Bac7(1–16) in their N-terminal sequences in complex with bacterial ribosomes, the first six residues of Bac7 form a domain different from the structure of the N-terminus of Onc112. This compact domain, in which Arg residues form an arranged positively charged structure, makes a significant contribution to the interaction of Bac7 with the ribosome. Moreover, the N-terminal RRIR motif also plays an important role in the internalization of Bac7(1–35) into bacterial cells [12].

PA-AMPs are synthesized in organisms in the form of inactive precursors that undergo proteolytic cleavage to form active AMPs [5,7,12]. PA-AMPs, including Bac7 and Onc112, mainly act on Gram-negative bacteria [7]. This specificity is determined by the fact that AMPs penetrate bacterial cells through the inner-membrane transporter proteins, such as SbmA and the YjiL-MdtM transporter system [10,12,24,25]. Positively charged residues distributed along the PA-AMP chain are believed to be necessary for their effective uptake by the cell via the SbmA transporter [26].

PA-AMPs exhibit low toxicity in eukaryotes [27,28]. On the other hand, Bac(1–35) was shown to inhibit eukaryotic translation in vitro [10]. The absence of a toxic effect on mammalian cells is explained by its complex mechanism of proteolytic activation and internalization via an endocytotic process that minimizes contact with the mammalian cytosolic ribosomes.

The appearance of bacterial resistance to PA-AMPs is not often observed. However, resistance can occur in the case of the mutation or deletion of the transporter protein genes required for the entry of PA-AMPs into bacterial cells, in particular, *sbmA* [7,24,25,29]. Another type of bacterial resistance, due to ribosomal mutations, was observed in Onc112 [14]. The nucleotide substitutions A2503C and A2059C in the 23S RNA sequence, and especially the double substitution A2503C/A2059G, increased *E. coli*’s resistance to Onc112 but not to Bac7(1–35).

In recent years, the search has continued for derivatives of both oncocin and bactenecins that are resistant to proteolytic degradation, have broader antibacterial activity, act against resistant bacterial strains, and possess other interesting and useful properties. These kinds of derivatives have been created either by replacing various amino acid residues with others contributing to the stability against proteolysis [30,31,32] or by combining peptides related to Bac7 with other antibiotics [33], antiviral drugs [34], or peptide nucleic acids (PNAs) [35] within the same molecule, or by conjugating two different AMPs [36], as well as on the basis of deep mutational scanning [37].

For some ribosome-targeting antibiotics, it has been demonstrated that their modification by alkyl-TPP groups leads to improved penetration into bacterial cells and cancer mitochondria, enhanced activity, and increased affinity for ribosomes [38,39,40,41]. In addition, it has also been shown that alkyl-TPP itself exhibits antibacterial properties [42,43].

Herein, we continue the research on rationally designed synthetic antimicrobial compounds that combine two different pharmacophores within a single molecule [38,39,44]. A new series based on a decapeptide related to the sequences of Bac7 and Onc112 and alkyl-TPP cations (Figure 1) was created in order to expand the spectrum of the antimicrobial activity of the AMP and provide new beneficial properties to the analogs. Using various biochemical and microbiological assays, we showed which properties were preserved or changed in the new derivatives compared were the original compounds, as well as what new properties appeared that might be valuable for the creation of new antibacterial agents.

**Figure 1 pharmaceutics-16-00148-f001:**
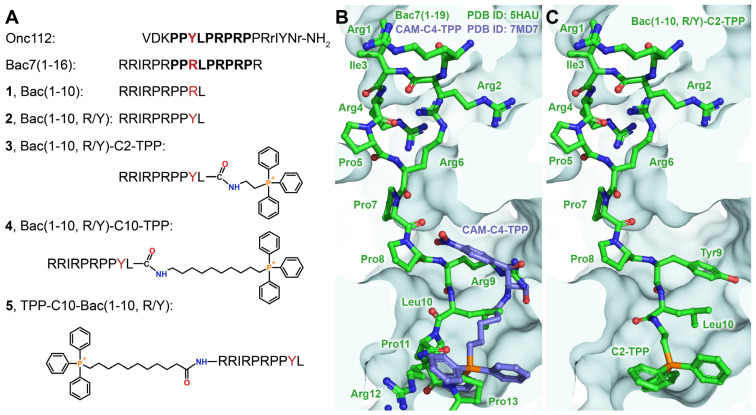
Structures of TPP analogs of Bac7. (**A**) Structures and abbreviations of the Bac7 analogs obtained in this work. The sequence that differs by only one amino acid, Tyr in the case of Onc112 and Arg in the case of Bac7(1–16), is shown in bold. (**B**) Superimposition of the crystal structures of Bac7(1–19) (green, PDB ID: 5HAU) [14] and TPP analog of chloramphenicol (blue, CAM-C4-TPP, PDB ID: 7MD7) [38] in complexes with the *Thermus thermophilus* ribosomes. (**C**) Model of the complex of compound **3** (Bac(1–10, R/Y)-C2-TPP) with the bacterial ribosome, derived from superimposition of X-ray diffraction data.

## 2. Materials and Methods

### 2.1. Chemicals and Materials

The following reagents and solvents were used: amino acids, their derivatives, and 2-chlorotrityl chloride resin for solid-phase peptide synthesis (2CTC Resin), HATU (O-(7-azabenzotriazol-1-yl)-N,N,N′,N′-tetramethyluronium hexafluorophosphate), and TIS (triisopropylsilane) (Iris Biotech, Marktredwitz, Germany and GL Biochem Shanghai, Shanghai, China); TentaGel HL NH_2_ resin (Rapp Polymere, Tubingen, Germany); HBTU (hexafluorophosphate benzotriazole tetramethyl uronium), DIPEA (N,N-diisopropylethylamine), triphenylphosphine, 1,2-dibromoethane, 1,10-dibromodecane, 11-bromoundecanoic acid, and 7N NH_3_ in methanol (Sigma-Aldrich, Steinheim, Germany); 4-methylpiperidine (Mosinter Chemicals, Ningbo, China); BODIPY-FL-C3-OSu (Lumiprobe, Moscow, Russia); CH_2_Cl_2_, dimethylformamide (DMF), and trifluoroacetic acid (TFA) (PanReac AppliChem, Darmstadt, Germany, or Solvay S.A., Brussels, Belgium); acetonitrile (CH_3_CN) (Biosolve Chimie, Dieuze, France); hexafluoro-2-propanol (HFIP) (Acros Organics, Geel, Belgium); and BODIPY-ERY, the fluorescent erythromycin derivative, which was synthesized as described previously in [45].

### 2.2. Chromatography

TLC was performed on Kieselgel 60 F254 plates (Merck, Darmstadt, Germany). For column chromatography, silica gel 60 (0.063–0.200 and 0.04–0.063 mm, Macherey-Nagel, Dueren, Germany) was used. Compounds containing UV-absorbing groups were detected with a Camag UV cabinet (Camag, Muttenz, Switzerland). Substances containing amino groups were visualized with a ninhydrin reagent. TPP and its derivatives were detected using Dragendorff’s reagent.

HPLC was performed using the Agilent 1200 system (a vacuum degasser, quaternary pump, autosampler, and diode array detector) (Santa Clara, CA, USA) equipped with a 4.6 × 250 mm Luna C18(2) column (5 μm) (Phenomenex, Torrance, CA, USA) with a flow rate of 1.0 mL/min, a temperature of 25 °C, UV detection at 220 nm, a gradient of 5–30% B for 12 min, 30–80% B for 3 min (peptides **1** and **2**), and 20–95% B for 20 min (TPP derivatives **3**–**5**) (A: 0.05% aqueous TFA; B: 0.05% TFA in CH_3_CN). Preparative HPLC was performed using the Knauer semi-preparative system Smartline (Knauer, Berlin, Germany) equipped with Smartline 1050 HPLC pumps, a Smartline UV detector 2520, a Smartline Manager 5050, and a Beckman Coulter Ultrasphere ODS column (5 microns, 250 × 10 mm, Brea, CA, USA). ClarityChrom 8.1 software was applied. The following conditions were used: a flow rate of 5 mL/min, ambient temperature, and elution with an appropriate gradient of CH_3_CN in 0.05% TFA. Preparative purification of Onc112 was carried out in a Gilson HPLC system (a 333/334 pump with a 215 liquid handler, Lewis Center, OH, USA) equipped with a YMC Triart C18 (150 × 30 mm) column and a UV detector at 210 and 280 nm. Onc112 was eluted in an aqueous gradient of CH_3_CN (from 10 to 55% for 30 min) with 0.1% TFA at a flow rate of 70 mL/min.

### 2.3. Liquid Chromatography–Mass Spectrometry

Liquid chromatography–mass spectrometry was performed using a UPLC/MS/MS system consisting of an Acquity UPLC chromatograph (Waters Corporation, Milford, MA, USA) and a TQD quadrupole mass spectrometer (Waters Corporation, Milford, MA, USA) with the registration of positive ions carried out using the ESI-MS method with an Acquity BEH column C18 (1.7 microns, 50 × 2.1 mm; Waters Corporation, Milford, MA, USA), a flow rate of 0.5 mL/min, a temperature of 35 °C, and elution with a gradient of 5–100% CH_3_CN in 20 mM of HCOOH for 4 min. UPLC-MS analysis of Onc112 was performed using a Thermo Finnigan LCQ Deca XP ion trap spectrometer (Thermo Fisher Scientific, Waltham, MA, USA) with the Thermo Accela UPLC system (Thermo Fisher Scientific, Waltham, MA, USA) equipped with an Atlantis T3 C18 (150 × 2 mm) column (Waters Corporation, Milford, MA, USA).

### 2.4. Mass Spectrometry

MALDI-TOF mass spectra were recorded with an UltrafleXtreme MALDI-TOF mass spectrometer (Bruker Daltonics, Bremen, Germany) equipped with a UV laser (Nd) in the reflectron positive-ion mode.

High-resolution mass spectra (HRMS) were recorded with an Orbitrap Elite Hybrid mass spectrometer (Thermo Fisher Scientific, Waltham, MA, USA) equipped with an electrospray ionization (ESI) source in positive-ion mode. The detailed HRMS data for compounds **1**–**5** are presented in the Appendix A.

### 2.5. ^1^H and ^13^C NMR

^1^H and ^13^C NMR spectra were recorded with a Bruker Avance spectrometer (Bruker, Bremen, Germany) with an operating frequency of 400 MHz for ^1^H, 101 MHz for ^13^C, and 162 MHz for ^31^P at 298 K in DMSO-d_6_ using tetramethylsilane as an internal reference. The spectra were processed and analyzed using Mnova 12.0 software (Mestrelab Research, Santiago de Compostela, Spain).

### 2.6. Molecular Modeling

The molecular design was performed by means of static modeling using Avogadro 1.1.1 [46] and PyMOL 2.6 software (The PyMOL Molecular Graphics System, Version 2.6 Schrödinger LLC, Portland, OR, USA).

### 2.7. Synthetic Procedures

The scheme for the synthesis of peptides **1** and **2** and their TPP derivatives **3**–**5** is represented in Appendix A. NH_2_-Cn-TPP (*n* = 2, 10) was obtained in two stages from TPP via the conjugation of TPP with dibromoalkanes at 85 °C for 72 h, according to [47], and the amination of the resulting bromoalkyl-TPP derivatives with 7M ammonia in methanol at 85 °C for 4 h [48]. TPP-C10-COOH was synthesized from TPP and 11-bromoundecanoic acid as described in [49].

Peptides **1** and **2**, and analog **5** were synthesized using the standard Fmoc solid-phase peptide synthesis protocol using 2-chlorotrityl chloride resin and HBTU/DIPEA activation [50,51,52]. The synthesis of Onc112 was carried out using a custom-made automated parallel peptide synthesizer. The Fmoc strategy with HATU/DIPEA activation was applied. Compounds **3** and **4** were obtained via the conjugation of the corresponding protected peptide with NH_2_-Cn-TPP (*n* = 2, 10) using HBTU as an activating agent. BODIPY-Bac(1–10) was synthesized from peptide **1** and BODIPY-FL-C3 succinimidyl ester (see the Appendix A for more detailed information on the synthetic procedures, HRMS data (Appendix A and Appendix A), and HPLC analysis data (Appendix A)).

***1****: Bac*(*1–10*). HPLC: t_R_ = 10.25 min (gradient of 5–30% B for 12 min and 30–80% B for 3 min). MALDI-TOF MS: *m*/*z* calculated for [C_57_H_105_N_25_O_11_+H]^+^—1316.8; found—1316.8. HRMS: *m*/*z* calculated for [C_57_H_105_N_25_O_11_+2H]^2+^—658.9286; found—658.9285.

***2****: Bac*(*1–10*, *R*/*Y*). HPLC: t_R_ = 10.99 min (gradient of 5–30% B for 12 min and 30–80% B for 3 min). MALDI-TOF MS: *m*/*z* calculated for [C_60_H_102_N_22_O_12_+H]^+^—1323.8; found—1323.7. HRMS: *m*/*z* calculated for [C_60_H_102_N_22_O_12_+2H]^2+^—662.4097; found—662.4100.

***3****: Bac*(*1–10*, *R*/*Y*)*-C2-TPP.* HPLC: t_R_ = 6.80 min (gradient of 20–95% B for 20 min). MALDI-TOF MS: *m*/*z* calculated for [C_80_H_121_N_23_O_11_P]^+^—1610.9; found—1610.9. HRMS: *m*/*z* calculated for [C_80_H_121_N_23_O_11_P+H]^2+^—805.9710; found—805.9708.

***4****: Bac*(*1–10*, *R*/*Y*)*-C10-TPP.* HPLC: t_R_ = 8.91 min (gradient of 20–95% B for 20 min). MALDI-TOF MS: *m*/*z* calculated for [C_88_H_137_N_23_O_11_P]^+^—1723.1; found—1723.0. HRMS: *m*/*z* calculated for [C_88_H_137_N_23_O_11_P+2H]^3+^—575.0249; found—575.0249.

***5****: TPP-C10-Bac*(*1–10, R*/*Y*). HPLC: t_R_ = 8.47 min (gradient of 20–95% B for 20 min). MALDI-TOF MS: *m*/*z* calculated for [C_89_H_136_N_22_O_13_P]^+^—1752.0; found—1751.9. HRMS: *m*/*z* calculated for [C_89_H_136_N_22_O_13_P+H]^2+^—877.0248; found—877.0250.

*BODIPY-Bac*(*1–10*). Fluorescence (MeOH): λ_ex_ = 505 nm; λ_em_ = 510 nm. LC-MS: *m*/*z* calculated for [C_71_H_118_BF_2_N_27_O_12_+2H]^2+^—795.98; found—794.82. t_R_ (UPLC) = 0.89 min. MALDI-TOF MS: *m*/*z* calculated for [C_71_H_118_BF_2_N_27_O_12_+H]^+^—1591.0; found—1591.0.

*Onc112.* LC-MS: t_R_ = 5.78 min (gradient of 5–55% 0.1% A aqueous TFA in 0.1% TFA in CH_3_CN for 17 min); *m*/*z* calculated for [C_109_H_177_N_37_O_24_+H]^+^—2390.4; found—2390.8.

### 2.8. In Vitro Binding Assay

70S ribosomes were isolated from *E. coli* MRE600 cells according to a published procedure [53]. The binding of BODIPY-Bac(1–10) to 70S *E. coli* ribosomes was performed according to the following procedure. BODIPY-Bac(1–10) (16 nM) was incubated with ribosomes (from 0.5 nM to 2000 nM) for 2 h at 26 °C in a buffer containing 20 mM of HEPES-KOH (pH 7.5), 50 mM of NH_4_Cl, 10 mM of Mg(CH_3_COO)_2_, 4 mM of β-mercaptoethanol, and 0.05% Tween-20. The binding affinities of peptides **1** and **2**, their TPP analogs **3**–**5**, as well as the control compounds for the *E. coli* ribosome were analyzed via a competition-binding assay using fluorescently labeled BODIPY-Bac(1–10) or BODIPY-ERY, as described before in [38,54]. The fluorescent compound (16 nM) was mixed with ribosomes (84 nM for BODIPY-Bac(1–10) and 46 nM for BODIPY-ERY) in the buffer. Solutions of the tested compounds were added to the obtained complexes to final concentrations ranging from 0.01 to 100 µM. The mixtures were incubated for 2 h at 26 °C, and then the values of fluorescence anisotropy were measured with a VICTOR X5 Multilabel Plate Reader (PerkinElmer, Waltham, MA, USA) using a 384-well plate. The excitation wavelength was 485 nm, and the emission wavelength was 535 nm. From the data obtained, apparent dissociation constants were calculated [55].

### 2.9. In Vitro Translation Inhibition Assays

The inhibition of firefly luciferase synthesis by the tested compounds was assessed in vitro, as indicated below. Briefly, in vitro-transcribed firefly luciferase mRNA (*fluc*) was translated using the *E. coli* S30 Extract System for Linear Templates (Promega, Madison, WI, USA). Reaction mixtures (with a 5 μL total volume) supplemented with 0.1 mM of a mixture of all canonical amino acids, 4 U of RiboLock RNase Inhibitor (Thermo Fisher Scientific, Waltham, MA, USA), and 0.1 mM of D-luciferin (Sigma-Aldrich, Burlington, MA, USA) were pre-incubated at RT for 5 min after the addition of the tested compounds at a final concentration of 30 µM. Then, 50 ng of the mRNA was added to each reaction tube, and the mixtures were immediately subjected to continuous chemiluminescence measurements using the VICTOR X5 Multilabel Plate Reader (PerkinElmer, Waltham, MA, USA) at 37 °C for 30 min. The maximal rates of the firefly luciferase (Fluc) accumulation were calculated using the OriginPro 7.5 software. The values were normalized to a positive control (0.3% DMSO, assigned a value of 100%).

The inhibition of eukaryotic translation was measured in a lysate of HEK293T cells, as described previously in [44], with minor modifications. Briefly, the reaction was carried out in a total volume of 10 µL with the following reagents: 5 µL of HEK293T cell lysate, prepared as described in [56]; 1 µL of a translation buffer 10× (200 mM of HEPES-KOH (pH 7.6), 10 mM of DTT, 5 mM of spermidine-HCl, 80 mM of creatine phosphate, 10 mM of ATP, 2 mM of GTP, and 0.25 mM of each amino acid); 0.5 µL of potassium acetate (2 M); 0.5 µL of magnesium acetate (20 mM); 0.5 µL of D-luciferin (10 mM) (Sigma-Aldrich, Burlington, MA, USA); 0.05 µL of RiboLock RNase Inhibitor (40 U/µL) (Thermo Fisher Scientific, Waltham, MA, USA); 0.5 µL of nuclease-free water; 1 µL of the tested compound or nuclease-free water; and 1 µL of Fluc mRNA (100 ng), capped and polyadenylated. All compounds were tested at a final concentration of 30 μM. Reactions containing all components, except mRNA, were pre-incubated at 30 °C for 5 min. Then, the mRNA was added, and the reactions were incubated in a CLARIOstar^®^ Plus Microplate Reader (BMG Labtech, Ortenberg, Germany) at 30 °C for 1.5 h with continuous measurement of the luciferase activity. Maximal rates of the luciferase accumulation were calculated using the OriginPro 7.5 software. The values were normalized to a positive control (0.3% DMSO, assigned a value of 100%).

### 2.10. Bacterial Inhibition Assays

#### 2.10.1. Detection of Translation Inhibitors Using the pDualrep2 Reporter Strain

The “pDualrep2” system was used to evaluate the mechanism of the antimicrobial action of the synthesized compounds. This system is based on the hypersensitive strain *E. coli* JW5503 (∆*tolC*) (KanS) [57] transformed with the “pDualrep2” plasmid, which allows one to sort out suppressors of protein synthesis or SOS response inducers [58]. Briefly, 1 µL of 10 mM water solutions of Bac(1–10), Bac(1–10, R/Y), Bac(1–10, R/Y)-Cn-TPP, TPP-C10-Bac(1–10, R/Y), and Onc112 were applied onto the agar plate that already contained a layer of the reporter strain *E. coli* JW5503 (∆*tolC*) (KanS) pDualrep2, and then the plate was incubated overnight at 37 °C. The agar plate was scanned using the ChemiDoc^TM^ Imaging System (Bio-Rad Laboratories, Hercules, CA, USA), containing two channels, “Cy3-blot” (553/574 nm; green pseudocolor) for Turbo Red Fluorescent Protein (TurboRFP) detection and “Cy5-blot” (588/633 nm; red pseudocolor) for Katushka2S fluorescence. Translation inhibitors trigger the induction of Katushka2S expression, while TurboRFP is upregulated by the SOS response. Levofloxacin (LEV; 50 μg/mL, 1 μL) and erythromycin (ERY; 5 mg/mL, 1 μL) were utilized as positive controls for DNA and protein synthesis inhibitors, respectively.

#### 2.10.2. Antibacterial Activity of Substances on Agar Plates

Antibiotic activity was also evaluated against the following strains: *E. coli* JW5503 Δ*tolC* (KanS) *ermC*, modified with the plasmid pKH80 [59], providing resistance to erythromycin due to the expression of ErmC methyltransferase; *E. coli* SQ171 Δ*tolC* transformed with the pAM552 plasmid [60]; *E. coli* SQ171 Δ*tolC* modified with the pAM552 plasmid with A2059G substitution in the 23S rRNA; *E. coli* JW1052 Δ*mdtH*; *E. coli* JW0912 Δ*ompF*; and *E. coli* JW0368 Δ*sbmA* [57]. The procedure was performed as described previously in [33]. In short, Petri dishes filled with LB solid medium containing a selective antibiotic and 1.5% agar were covered with the tested strains. After that, 1 µL of 10 mM water solutions of Bac(1–10), Bac(1–10, R/Y), Bac(1–10, R/Y)-Cn-TPP, TPP-C10-Bac(1–10, R/Y), and Onc112, LEV (25 μg/mL) in water, and ERY (5 mg/mL and 50 mg/mL) in DMSO were applied onto the agar plates. After overnight incubation at 37 °C, the plates were scanned using the ChemiDoc^TM^ Imagining System (Bio-Rad Laboratories, Hercules, CA, USA) in three channels (Cy2, Cy3, and Cy5). The obtained images were processed in the Image Lab^TM^ software (Version 6.0.1, Bio-Rad).

#### 2.10.3. Minimum Inhibitory Concentration (MIC) Determination

The MIC values for Bac(1–10), Bac(1–10, R/Y), Bac(1–10, R/Y)-Cn-TPP, TPP-C10-Bac(1–10, R/Y), and Onc112 were determined via LB broth microdilution, as recommended by CLSI in the *Methods for Dilution Antimicrobial Susceptibility Tests for Bacteria that Grow Aerobically*, Approved Standard, 9th ed., CLSI document M07-A9, using in-house-prepared panels. The compounds were diluted in a 96-well microtiter plate to final concentrations ranging from 0.1 to 100 µM in a 200 µL aliquot of the bacterial suspension, followed by incubation at 37 °C for 18 h. The cell concentration was estimated according to the absorbance (A600). The measurements were performed with a VICTOR X5 Multilabel Plate Reader (PerkinElmer, Waltham, MA, USA). The following bacterial strains were used: *E. coli* BW25113, *E. coli* JW0368 Δ*sbmA*, *B. subtilis* 168, and *B. subtilis* CFR [39]. The lowest concentration of a tested compound that completely inhibited bacterial growth was considered the MIC.

### 2.11. Determination of Membrane Potential in B. subtilis

The membrane potential of *B. subtilis* was assessed by measuring the fluorescence emitted by the potential-dependent probe, diS-C3-(5) [61]. Bacterial cells derived from the overnight culture of *B. subtilis* were inoculated into a fresh LB medium and allowed to grow for 24 h until reaching an optical density of 0.8 at 600 nm. Subsequently, the bacteria were subjected to a 20-fold dilution in a buffer comprising 100 mM of KCl and 10 mM of Tris, adjusted to a pH of 7.4. Fluorescence measurements were conducted at 670 nm (with excitation at 630 nm) using a Fluorat-02-Panorama fluorimeter manufactured by Lumex Instruments in St. Petersburg, Russia.

### 2.12. In Vitro Survival Assay (MTT Assay)

The cytotoxicity of the substances under study was tested using the MTT (3-(4,5-dimethylthiazol-2-yl)-2,5-diphenyltetrazolium bromide) assay [62] with some modifications [39,44]. The detailed procedure for the MTT assay is given in the Appendix A (Appendix A).

## 3. Results and Discussion

### 3.1. Modeling of TPP Analogs of Bac7

When designing analogs whose action is directed at bacterial ribosomes, we decided to use as short a peptide as possible from the Bac7 N-terminal sequence and a TPP cation attached to the peptide via linkers of various lengths (Figure 1A). It is known that the contacts formed by amino acid residues belonging to the N-terminal peptide sequence make the greatest contribution to the binding of Bac7, as well as Onc112, to bacterial ribosomes. Almost identical conformations of the peptide backbone and side chains are observed for residues 7–13 of Bac7(1–16) and residues 4–10 of Onc112, with the Arg9 of Bac7(1–16) substituting for the Tyr6 of Onc112 [10,11,12]. Previous studies have shown that the introduction of a TPP group into the structure of chloramphenicol amine significantly increases its affinity for bacterial ribosomes [38,39]. Thus, the superimposition of the crystal structures of Bac7(1–19) (PDB ID: 5HAU) [14] and the TPP analog of chloramphenicol (CAM-C4-TPP; PDB ID: 7MD7) [38] in complexes with *Thermus thermophilus* ribosomes was applied in order to model the structures of new TPP analogs (Figure 1B).

Therefore, a decapeptide (1–10) from the sequence of Bac7 (Figure 1A; Bac(1–10) (**1**)) was chosen as a short peptide that could ensure the proper accommodation of the TPP group upon conjugation. Moreover, the Arg9 residue was replaced with a Tyr one, which is located in a similar position in the structure of Onc112 upon binding to the ribosome (compound **2**—Bac(1–10, R/Y)). The alkyl-TPP cations should be attached to the C-terminal residue of peptide **2** (Leu10) so that upon binding to the ribosome, the TPP moieties are directed deep into the NPET. The length of the linker for compound **3** (Bac(1–10, R/Y)-C2-TPP) was chosen so that the TTP group would fall into the same site on a ribosome as in the case of CAM-C4-TPP (Figure 1C). The choice of the linker for compound **4** (Bac(1–10, R/Y)-C10-TPP) was also based on previous data on the ribosome binding and the antibacterial activity of the TPP analog of chloramphenicol containing a longer linker [39]. Such an alkyl-TPP moiety can contribute to the interaction of the analog with the ribosome and facilitate penetration into bacterial cells by reducing the membrane potential. For comparison, a similar compound containing a decyl-TPP moiety at the N-terminus of peptide **2** was also constructed (Figure 1A; TPP-C10-Bac(1–10, R/Y) (**5**)).

### 3.2. Synthesis of TPP Analogs of Bac7

Peptides **1** and **2** were synthesized via solid-phase synthesis on a 2CTC resin using the standard Fmoc/Pbf(tBu) procedure and HBTU as a condensation agent. The Fmoc protective group was removed with a piperidine solution. TFA along with scavengers (reagent K) was used for the cleavage of peptides from the resin and the removal of side-chain protective groups (Appendix A). For the preparation of peptide **6**, which served as a parent compound for the syntheses of analogs **3** and **4**, HFIP was used for cleavage from the resin. In the presence of this reagent, the Pbf and tBu protective groups remained intact (Appendix A). The C-terminal carboxyl group of peptide **6** was modified with amines, (2-aminoethyl)(triphenyl)phosphonium bromide or (10-aminodecyl)(triphenyl)phosphonium bromide (Appendix A). They were prepared in advance in two stages from dibromoalkanes and TPP: (i) the formation of bromoalkyl derivatives of TPP [47], and (ii) their amination with a 7M ammonia methanol solution [48]. Analog **5** was synthesized directly on the resin via the condensation of the carboxyl derivative of TPP [49] with a peptidyl polymer. The compounds were purified on silica gel or via HPLC and analyzed using the LC-MS, NMR, and HR-MS techniques (Appendix A).

### 3.3. TPP Derivatives of Decapeptide from N-Terminal Sequence of Bac7 Bind to the Bacterial Ribosome

Since TPP analogs of Bac(1–10) were created as compounds acting on bacterial ribosomes, their ability to bind to 70S *E. coli* ribosomes was studied. For this purpose, starting from peptide **1**, its fluorescent derivative BODIPY-Bac(1–10) was synthesized, and its binding to *E. coli* ribosomes was investigated (Appendix A). The apparent dissociation constant (K_D_) of BODIPY-Bac(1–10) was 77 ± 5 nM.

To assess the affinity of peptides **1** and **2** and analogs **3**–**5** for the 70S *E. coli* ribosome, a competition-binding assay with BODIPY-labeled decapeptide BODIPY-Bac(1–10) was used (Figure 2A). According to the results, the decapeptide (1–10) from the Bac sequence (**1**) retained the ability to bind to bacterial ribosomes (K_D_ = 1.4 ± 0.1 µM), although its affinity was significantly lower than that of Onc112 (K_D_ = 3 ± 1 nM; Appendix A) and Bac7(1–16) [12]. Replacing the Arg residue at position 9 with Tyr, which occupies the corresponding position in the Onc112 structure upon binding to a bacterial ribosome, slightly increased the affinity of peptide **2** (K_D_ = 1.0 ± 0.1 µM) compared with that of peptide **1**. The introduction of TPP moieties significantly improved the affinity of analogs **3**–**5** to bacterial ribosomes in comparison with the unmodified peptides **1** and **2**. Compounds **3** and **4** exhibited approximately the same affinity (K_D_ = 0.26 ± 0.03 µM and 0.25 ± 0.04 µM, respectively), while analog **5** showed the strongest binding (K_D_ = 0.16 ± 0.02 µM). Despite having a lower affinity for ribosomes compared with Onc112, the TPP derivatives still exhibited dissociation constants that are typical of numerous PA-AMPs [9,63,64].

Compounds **1**–**5** also displaced fluorescently labeled erythromycin BODIPY-ERY [54] from its binding site (Appendix A). This observation suggests that the compounds interact with the NPET, which contains overlapping binding sites for both ERY and Bac7 [14].

### 3.4. Bac(1–10, R/Y)-C2-TPP and Bac(1–10, R/Y)-C10-TPP Selectively Inhibit Prokaryotic Translation

The next step in testing antimicrobial compounds targeting the bacterial ribosome involves probing their ability to inhibit protein biosynthesis, which was assessed using the in vitro translation reaction in a cell-free transcription–translation system based on the *E. coli* S30 extract. Analogs **3** and **4** inhibited the bacterial translation of firefly luciferase (Fluc) mRNA almost completely, like Onc112, while peptides **1** and **2** practically did not inhibit this process, as well as analog **5**, which weakly inhibited bacterial translation (Figure 2B, white bars; Appendix A). These data confirm that analogs **3** and **4**, designed so that they bind to the ribosome in such a way that the TPP group is directed deep into the NPET, are able to inhibit bacterial translation, as was observed earlier for TPP and berberine analogs of chloramphenicol [38,39,44]. It is noteworthy that even though compound **5** exhibited strong binding to the ribosome, it showed very low inhibitory activity. This phenomenon has been observed among various derivatives of chloramphenicol, which bind to the upper part of the NPET and PTC [65].

It has been shown that Bac7(1–35) and Bac7(1–16), like some other Pro-Arg-rich AMPs, can affect not only the process of bacterial translation but also eukaryotic translation [12]. This property can be considered a disadvantage for antibacterial compounds since it can lead to various side effects when using these agents as medications. A cell-free translation system based on HEK293T lysates was applied to test how the TPP analogs of Bac(1–10) influence eukaryotic translation in vitro (Figure 2B, grey bars). At a concentration of 30 µM, compounds **1**–**5** inhibited translation much less than Onc112. Thus, the TPP analogs **3** and **4** were found to be more selective inhibitors of bacterial translation than Onc112 and Bac7 [12].

### 3.5. TPP Derivatives of Decapeptide Related to Bac7 and Onc112 Exhibit Antibacterial Activity against Various Strains

#### 3.5.1. Using the Double-Reporter System pDualrep2 for Preliminary Assessment of the Mechanism of Action

The JW5503 (∆*tolC*) (KanS) pDualrep2 strain was used for the determination of the antibacterial action of the obtained compounds, as well as the possible mechanisms of their antibacterial activity [58]. As described above, the *E. coli* ∆*tolC* reporter system is based on TurboRFP and Katushka2S, which are proteins whose fluorescence can be registered independently. This reporter system can be used for screening either protein synthesis or DNA replication inhibitors. The expression of the far-red fluorescent protein, Katushka2S, occurs in the pDualrep2 reporter system in the presence of ribosome-stalling compounds and is detected as a red pseudocolor (Appendix A, ERY). The induction of the expression of the red fluorescent protein, TurboRFP, by substances that cause the SOS response can also be detected as a green pseudocolor (Appendix A, LEV).

Bac(1–10, R/Y)-C2-TPP (**3**), Bac(1–10, R/Y)-C10-TPP (**4**), and TPP-C10-Bac(1–10, R/Y) (**5**) demonstrated an inhibitory effect on the growth of bacterial cells in this test (Appendix A). On the contrary, peptides **1** and **2** did not exhibit any antibacterial activity (Appendix A), which is consistent with the data available in the literature [23]. In later experiments, either peptide **1** or peptide **2** were used as control compounds when checking the antibacterial activity of the synthesized analogs. For Bac(1–10, R/Y)-C2-TPP (**3**) and Bac(1–10, R/Y)-C10-TPP (**4**), as well as for Onc112 and positive controls, ERY, red pseudocolor reporter induction was observed, indicating that these compounds specifically inhibit protein synthesis. Although C10-TPP caused a large growth inhibition zone, it did not induce the reporters, which implies that it has antibacterial properties, but with a different mechanism of action that cannot be detected by the pDualrep2 reporter. Overall, the results on the reporter strain agree well with the data on the in vitro translation system (Figure 2B and Appendix A).

Thus, the mechanism of action of the TPP analogs of Bac(1–10, R/Y) **3** and **4** on bacterial cells is similar to that of the parent PA-AMPs Bac7 and Onc112.

#### 3.5.2. TPP Derivatives of Decapeptide from N-terminal Sequence of Bac7 Exhibit Antibacterial Activity against Various Strains including Some Resistant Laboratory Strains

Like most PA-AMPs, Onc112 and Bac7 inhibit the growth of Gram-negative bacteria [7,27], while alkyl-TPPs and their derivatives act on Gram-positive bacteria [42]. Since the structures of compounds **3**–**5** combine a decapeptide, which is related to both Bac7 and Onc112, as well as the alkyl-TPP fragment, the activity of these analogs against various Gram-negative and Gram-positive strains was tested (Figure 3, Table 1). According to the results obtained, decapeptides **1** and **2** did not inhibit bacterial growth, which is consistent with the known data that peptides containing less than 16 a.a. from the Bac7 sequence do not exhibit antibacterial activity [23].

Unlike peptides **1** and **2**, TPP derivatives **3**–**5** containing modifications at both the N- and C-terminus of peptide **2** were active against *E. coli* strains and also inhibited the growth of *B. subtilis* in contrast with Onc112 (Table 1). It is also worth noting that while analogs **4** and **5** contain a decyl-TPP fragment (C10-TPP), which itself has antibacterial properties, analog **3** contains ethyl-TPP, which does not have such activity [42].

It was shown that mutations in the 23S rRNA at the NPET, such as A2503C, A2059C, and especially the double mutation (A2503C/A2059G), led to increased resistance to Onc112, but not to Bac7(1–35) [5,12]. The authors of [12] explained this effect by the structural features of Onc112 binding in the NPET due to the stacking interaction between the Arg11 of Onc112 and the A2062 nucleotide, which is in close proximity to both A2503 and A2059 [66], while the Arg16 of Bac7 compensates for the loss of interaction with A2062 since it forms a π-stacking interaction with the His69 of the ribosomal protein uL4 [14]. As a result, any mutation in these nucleotides can affect the rotation of A2062 and, as a consequence, interfere with the binding of Onc112 more than in the case of Bac7(1–35). In the structures of analogs **3** and **4**, there is no amino acid residue corresponding to the position of Arg11 in Onc112, but they contain a bulky alkyl-TPP substituent directed toward nucleotides A2062, A2503, and A2059 upon binding to the ribosome. Therefore, the TPP analogs of Bac7 were tested on a chloramphenicol-resistant *B. subtilis* strain *(B. subtilis* CFR) harboring a plasmid encoding a methyltransferase (Cfr) that modifies the A2503 in the 23S rRNA (Table 1). In addition, the macrolide-resistant *E. coli* strains SQ171 Δ*tolC* (A2059G), in which the A2059 nucleotide is replaced with G2059 in the 23S rRNA gene (Figure 3B), and JW5503 (∆*tolC*) (KanS) *ermC*, harboring a plasmid encoding a methyltransferase (ErmC) that catalyzes the methylation of A2058 in the 23S rRNA, were used (Figure 3C).

Analogs **4** and **5**, containing decyl-TPP moieties at the C-terminus or N-terminus of the peptide, were active against a resistant *B. subtilis* strain (Table 1). Analog **3** also suppressed the growth of this strain somewhat more strongly than chloramphenicol.

In *E. coli* strains resistant to macrolides (Figure 3), analogs **3** and **4** containing modifications at the C-terminus showed bacterial-growth-inhibiting activity, in contrast with erythromycin. However, it should be noted that Onc112 in these experiments also showed activity against resistant *E. coli* bacteria comparable to that demonstrated in the wild type.

#### 3.5.3. Bac(1–10, R/Y)-C2-TPP Penetrates into the *E. coli* Cells via the SbmA Transporter Protein

PA-AMPs were shown to act on Gram-negative bacteria by penetrating through the inner membrane without destroying it using special protein transporters such as the SbmA protein [24,26,67]. *E. coli* strains with the deletion of the *sbmA* gene became less sensitive to Bac7(1–35) and Bac7(1–16) as well as to oncocins, including Onc112, compared with wild-type strains [23,25]. Onc112 also, but to a lesser extent, suppressed the growth of *E. coli* strains with the deletion of the *mdtH* gene (which encodes another transporter protein) [25]. Simultaneous deletion of both the *sbmA* and *mdtH* genes resulted in the appearance of an Onc112-resistant phenotype compared with a wild-type *E. coli* strain [25].

The TPP analogs of decapeptide Bac(1–10, R/Y) **3**–**5**, as well as peptides **1** and Onc112, were tested on three laboratory *E. coli* strains with deletions of either transporter genes (*sbmA* and *mdtH*) associated with the inner membrane or the porin gene (*ompF*) associated with the outer membrane (Figure 4, Table 1). The latter protein is responsible for the passive transport of small molecules, such as nutrients and antibiotics, into cells [68].

As follows from the results obtained, there was no significant difference in the growth inhibition of *E. coli* Δ*ompF*, *E. coli* Δ*mdtH*, and a wild-type strain upon treatment with synthetic analogs **3**–**5** and Onc112 (Figure 4). Only in the case of the *E. coli ΔsbmA* strain, a decrease in the antibacterial activity of Onc112 and analog **3** compared with the wild-type strain was observed (Figure 4B). In order to validate this effect, MIC values were determined for all compounds against these two strains (Table 1). The corresponding values for Onc112 and analog **3** differed by approximately eight and four times, respectively. In the case of analogs **4** and **5**, as well as C10-TPP, the MIC values were the same. Based on these results, the conclusion was made that analog **3** is transported via the *E. coli* membrane mainly with the participation of the transporter protein SbmA, similar to Onc112. Unlike analog **3**, analogs **4** and **5**, containing a decyl-TPP fragment in their structures, are transferred into bacterial cells similarly to C10-TPP, using a different mechanism peculiar to alkyl derivatives of TPP [42,69].

Thus, we demonstrated that TPP analogs of a decapeptide related to Bac7 and Onc112 have antimicrobial action and are able to act against both Gram-negative bacteria, similar to oncocin and bactenecin, and Gram-positive ones, like alkyl derivatives of TPP. Moreover, TPP derivatives **3**–**5** were shown to inhibit the growth of some resistant laboratory strains. In addition, the decapeptide, as part of its TPP derivative **3** containing a short alkyl linker, was transferred into *E. coli* cells with the participation of the transporter protein SbmA, while the transport of TPP derivative **4** with a longer alkyl linker turned out to be independent of SbmA.

### 3.6. Bac(1–10, R/Y)-Cn-TPP Cause a Decrease in the Membrane Potential of B. subtilis

Previously, it was shown that alkyl-TPP molecules, containing eight or more methylene groups, as well as their derivatives, showed the ability to reduce the membrane potential of bacteria [39,42,43]. On the other hand, for Bac7(1–16) analogs containing substitutions of some amino acid residues with a hydrophobic aromatic Trp, it has been shown that these analogs can directly penetrate the bacterial cell through the membrane independently of transport proteins [70]. Thus, the effect of TPP analogs of a decapeptide related to Bac7 and Onc112 **3**–**5**, as well as parent peptides **1** and **2,** on the bacterial membrane potential of *B. subtilis* was further examined. Gramicidin A, a channel-forming antibiotic, was used to cause the disappearance of the bacterial membrane potential. The fluorescent-potential-sensitive dye diS-C3-(5) was applied to measure the change in the membrane potential of *B. subtilis* [61].

Bac(1–10, R/Y)-C10-TPP (**4**) at a concentration of 1 µM caused a rapid decrease in the membrane potential of *B. subtilis* to a level similar to that observed in the presence of gramicidin A (Figure 5). A decrease in the membrane potential, to a somewhat lesser extent, was observed in the presence of TPP-C10-Bac(1–10, R/Y) (**5**) at a concentration of 2.5 µM. These effects were expected because both these compounds contain a decyl-TPP moiety in their structures. Moreover, in the case of Bac(1–10, R/Y)-C2-TPP (**3**), containing a short ethyl linker between the TPP and the C-terminus of the peptide, there was also a release of the fluorescent dye from cells, which indicated a decrease in the membrane potential of *B. subtilis.* It is possible that the combination of several positive charges and a hydrophobic TPP group in this amphiphilic compound with a linear flexible structure might contribute to its interaction with negatively charged lipids (in particular, cardiolipin) in the *B. subtilis* membrane and promote peptide accumulation on the bacterial surface [71,72]. This also does not exclude the possible interaction of compound **3** with endogenous fatty acids, as happens in the case of the penetration of alkyl-TPP derivatives through the mitochondrial membrane [69]. Peptides **1** and **2** did not affect the membrane potential, even at a concentration of 50 µM.

Thus, the action of the TPP analogs of decapeptide related to Bac7 and Onc112 on bacteria can be associated with both the inhibition of translation and the depolarization of the bacterial membrane, similar to what was previously shown for the TPP and berberine analogs of chloramphenicol [39,44]. Moreover, unlike compounds from our previously studied series of penetrating hydrophobic cations conjugated with chloramphenicol, in this series of TPP derivatives of decapeptide related to Bac7 and Onc112, all compounds, both containing long and short alkyl linkers between TPP and the peptide, had an effect on the bacterial membrane. An arginine-rich peptide in the structure of Bac(1–10, R/Y)-C2-TPP apparently contributes to the effect of the analog on the membrane potential; however, the mechanism of this action needs to be clarified and further investigated.

### 3.7. Bac(1–10, R/Y)-C2-TPP Is Non-Toxic for Mammalian Cells

It is known that alkyl derivatives of TPP can be toxic to mammalian cells. Both antibacterial and toxic effects are observed for TPP derivatives containing a relatively long alkyl radical [39,73]. To test the cytotoxicity of TPP analogs **3**–**5** to mammalian cells, the Mosmann (“MTT”) assay was used [62].

The results show that analog **3**, containing a short linker between the C-terminal carbonyl group of the peptide and TPP, turned out to be non-toxic for all cell lines used: HEK293T, MCF7, VA13, and A549 (Table 2). The toxicities of analogs **4** and **5** containing decyl-TPP fragments, both at the C-end and at the N-end of the peptide, were significantly reduced compared with C10-TPP.

The absence of a toxic effect (Table 2; Appendix A) at MIC concentrations in combination with an inhibitory effect on the bacterial translation process and the ability of the Bac(1–10, R/Y)-C2-TPP to suppress the growth of both Gram-negative and Gram-positive bacteria, including some resistant strains, are important properties in relation to the further use of the found structural and functional patterns to create antibacterial drugs.

## 4. Conclusions

In this study, it was shown that using a rational design based on short decapeptides with a hybrid PA-AMP sequence of the first type, antimicrobial agents can be created by modifying these peptides with alkyl-TPP. The resulting conjugates retain the necessary properties of the PA-AMP for biological activity as antimicrobials, such as the ability to interact with the target of the PA-AMP in the cell, the bacterial ribosome; to suppress the synthesis of bacterial proteins; and to penetrate Gram-negative bacteria like the PA-AMP. On the other hand, due to the presence of alkyl-TPP cations in their structures, these conjugates can inhibit the growth of Gram-positive bacteria, reducing the membrane potential of bacterial cells. At the same time, the new analogs have some new properties that are not inherent to the original compounds individually. These properties were demonstrated in the study with examples of synthetic TPP derivatives of a decapeptide related to the sequences of Bac7 and Onc112. It was shown that while reducing the length of the AMP to 10 amino acid residues, the affinity to bacterial ribosomes dramatically decreased, and the modification of the peptide with alkyl-TPP moieties led to an increase in the affinity. The TPP analogs, designed so that upon binding to the ribosome, the TPP group could be directed deep into the NPET, were able to inhibit bacterial translation more selectively than Onc112 and Bac7. The TPP analogs of the decapeptide related to Bac7 and Onc112 acted against Gram-negative bacteria, similar to oncocin and bactenecin, and against Gram-positive bacteria, similar to other alkyl derivatives of TPPs. Also, they inhibited the growth of some resistant laboratory strains. New synthetic conjugates can enter bacterial cells both with the participation of transport proteins and through penetration due to alkyl-TPP cations. It has been shown that Bac(1–10, R/Y)-C2-TPP, containing a short alkylamide linker between the decapeptide and TPP, is transferred into *E. coli* cells via the transporter protein SbmA. TPP derivatives of the decapeptide Bac(1–10, R/Y) containing a decylamide as well as an ethylamide linker caused *B. subtilis* membrane depolarization, similar to alkyl-TPP. The analog Bac(1–10, R/Y)-C2-TPP, which has been shown to combine the antibacterial properties of alkyl-TPP and PA-AMP and to differ in terms of reduced side effects inherent to the parent molecules, such as toxicity and effects on eukaryotic translation, stands out in the series. These properties follow from the structure of the compound, which combines a short peptide related to the sequences of Bac7 and Onc112 with TPP, attached to the C-terminal amino acid residue of the peptide via a short alkylamide linker. The structural features of the TPP analogs of the short peptide related to Bac7 and Onc112 found in this study can be applied in the future for the creation of therapeutic antibacterial agents based on AMPs.

## Figures and Tables

**Figure 2 pharmaceutics-16-00148-f002:**
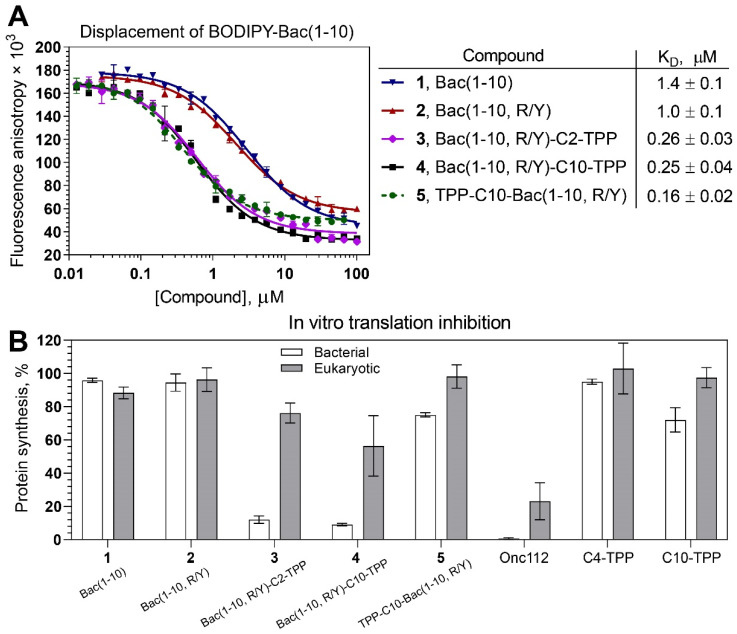
Binding affinity to bacterial ribosomes and the inhibition of protein synthesis by TPP analogs of short peptide from Bac7. (**A**) A competitive binding assay to test the affinity of peptides Bac(1–10) (**1**) and Bac(1–10, R/Y) (**2**) and their TPP analogs Bac(1–10, R/Y)-Cn-TPP (**3**: *n* = 2; **4**: *n* = 10) and TPP-C10-Bac(1–10, R/Y) (**5**) to *E. coli* 70S ribosomes measured using fluorescence anisotropy of fluorescently labeled peptide, BODIPY-Bac(1–10). All experiments were performed at least twice, and the error bars are the SD. The table displays the average apparent dissociation constants (K_D_) along with their CI (α = 0.05). (**B**) The inhibition of protein synthesis in vitro in the presence of Bac(1–10) (**1**), Bac(1–10, R/Y) (**2**), and TPP analogs **3**–**5** in cell-free bacterial (white bars) and eukaryotic (grey bars) systems. Onc112 and alkyl-TPPs (C4-TPP and C10-TPP) were used as controls. All compounds were tested at a final concentration of 30 μM. Relative maximal rates of the firefly luciferase (Fluc) accumulation in vitro are shown. Experiments were performed at least two times, and the error bars represent the SD.

**Figure 3 pharmaceutics-16-00148-f003:**
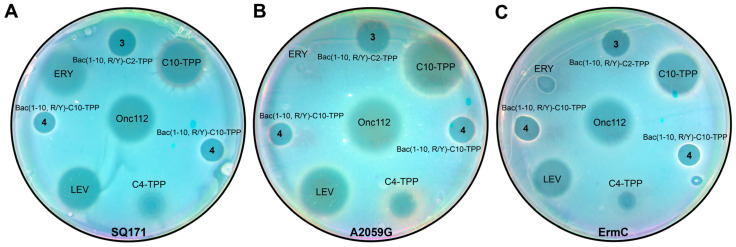
Antibacterial activity of TPP derivatives of decapeptide related to Bac7 and Onc112. Onc112, erythromycin (ERY), C10-TPP, and C4-TPP were used as controls. Activity of Bac(1–10, R/Y)-C2-TPP (**3**) and Bac(1–10, R/Y)-C10-TPP (**4**) was assessed using the *E. coli* Δ*tolC* strain SQ171 (**A**) and macrolide-resistant *E. coli* strains, SQ171 Δ*tolC* (A2059G), in which the A2059 nucleotide was replaced with G2059 in the 23S rRNA gene (**B**), and JW5503 (∆*tolC*) (KanS) *ermC*, harboring a plasmid encoding a methyltransferase, which catalyzes the methylation of A2058 in the 23S rRNA (**C**).

**Figure 4 pharmaceutics-16-00148-f004:**
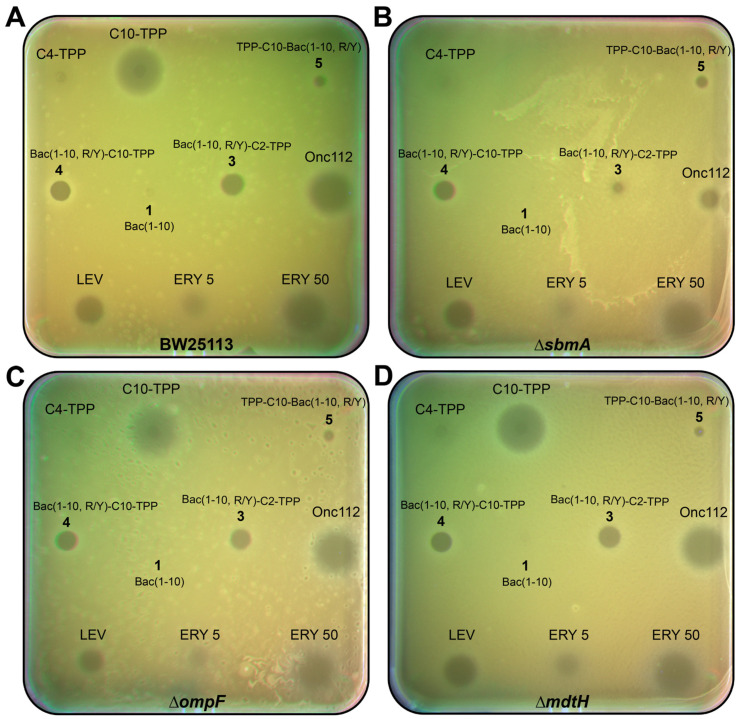
Antibacterial activity of TPP derivatives of decapeptide related to Bac7 and Onc112. Onc112, levofloxacin (LEV), erythromycin (ERY), C10-TPP, and C4-TPP were used as controls. Activity of Bac(1–10) (**1**), Bac(1–10, R/Y)-C2-TPP (**3**), Bac(1–10, R/Y)-C10-TPP (**4**), and TPP-C10-Bac(1–10, R/Y) (**5**) was assessed using the *E. coli* strain BW25113 (**A**) and *E. coli* strains BW25113 with the deletion of the following genes—*sbmA* (**B**), *ompF* (**C**), and *mdtH* (**D**)—associated with membrane transport.

**Figure 5 pharmaceutics-16-00148-f005:**
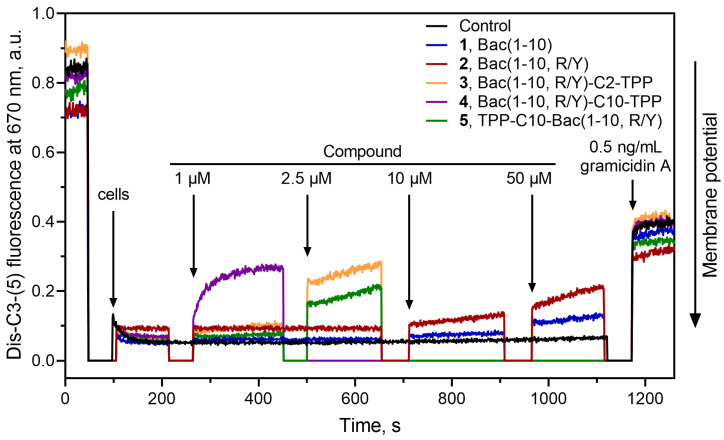
Effect of triphenylphosphonium analogs Bac(1–10, R/Y)-Cn-TPP (**3**: *n* = 2; **4**: *n* = 10), TPP-C10-Bac(1–10, R/Y) (**5**), and the parent peptides **1** and **2** on the kinetics of *B. subtilis* cell membrane potential measured using the fluorescent probe diS-C3-(5). The dye was evenly distributed in the medium, and fluorescence was observed. When cells are added, the dye accumulates inside living cells with potential, the dye self-quenches, and a decrease in fluorescence is observed. When substances that cause a decrease in membrane potential are added to cells, the dye is released into the medium, which causes an increase in fluorescence. Arrows indicate the points at which the respective compounds were introduced. Gramicidin A was employed as a control at a concentration of 0.5 ng/mL.

**Table 1 pharmaceutics-16-00148-t001:** Suppression of bacterial growth by TPP analogs of decapeptide related to Bac7 and Onc112. The following strains were used: *E. coli* BW25113, *E. coli* strain with the deletion of the *sbmA* gene (∆*sbmA*), *B. subtilis* 168, and chloramphenicol-resistant *B. subtilis* strain harboring the *cfr* gene (*B. subtilis* CFR). The values of the minimum inhibitory concentration (MIC, in µM) are shown ^1^.

	*E. coli* BW25113	*E. coli* ∆*sbmA*	*B. subtilis* 168	*B. subtilis*-CFR
Bac(1–10) (**1**)	-	-	>50	>50
Bac(1–10, R/Y) (**2**)	>100	>100	-	-
Bac(1–10, R/Y)-C2-TPP (**3**)	26.3	>100	12.5	50
Bac(1–10, R/Y)-C10-TPP (**4**)	100	100	1.6	0.8
TPP-C10-Bac(1–10, R/Y) (**5**)	5.3	5.3	12.5	12.5
Onc112	11.6	>90	>50	>50
C4-TPP	>100	>100	>100	>100
C10-TPP	50	50	1.6	0.8
ERY	170	170	<0.1	<0.1
Chloramphenicol ^2^	-	-	12	90

^1^ The MIC values were determined using the double-dilution method. The MIC for each compound was determined in triplicate in two independent sets. ^2^ Data are given according to the article in [39].

**Table 2 pharmaceutics-16-00148-t002:** Reduction in viability of human cell lines treated with TPP derivatives of decapeptide related to Bac7 and Onc112 (**3**–**5**) and the parent peptide Bac(1–10, R/Y) (**2**) measured via MTT assay. Concentrations that caused 50% inhibition of viability of cells (IC_50_, in µM) are shown (concentration–viability dependencies are given in Appendix A).

	HEK293T	MCF7	VA13	A549
Bac(1–10, R/Y) (**2**)	>50	>20	>50	>50
Bac(1–10, R/Y)-C2-TPP (**3**)	>50	>20	>50	>50
Bac(1–10, R/Y)-C10-TPP (**4**)	3.1 ± 0.5	10 ± 1	12 ± 1	9.2 ± 0.9
TPP-C10-Bac(1–10, R/Y) (**5**)	36 ± 4	>20	>50	>50
Onc112	>50	>20	>50	>50
C4-TPP	2.9 ± 0.5	8 ± 2	11 ± 2	6.2 ± 0.9
C10-TPP	<0.16	0.25 ± 0.07	<0.16	<0.16
Doxorubicin	<1.6	1.7 ± 0.2	1.2 ± 0.3	1.2 ± 0.1

## Data Availability

The data presented in this study are available in this article and Appendix A and can also be shared upon request.

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
