# Peer review of "Triphenylphosphonium Analogs of Short Peptide Related to Bactenecin 7 and Oncocin 112 as Antimicrobial Agents"

_pharmaceutics, 2024, doi:10.3390/pharmaceutics16010148_

Round 1

Reviewer 1 Report

Comments and Suggestions for Authors

The submitted manuscript demonstrated triphenylphosphonium analogues of short peptides related to bactenecin 7 and oncocin 112 as antimicrobial agents. It is a very well-designed project. However, there are a few comments that need to be addressed.

1.     Page 5, they should also provide the HPLC characterisation of their peptides in the supporting information.

2.     In the introduction, a couple of landmark summaries of Proline Rich AMPs are missing, such as CMLS, Cell. Mol. Life Sci. 59, 1138–1150 (2002). https://doi.org/10.1007/s00018-002-8493-8, Amino Acids 46, 2287–2294 (2014). https://doi.org/10.1007/s00726-014-1820-1

Comments on the Quality of English Language

See comments

Author Response

We are grateful to the Reviewers for the careful reading of our manuscript, as well as for the valuable comments made. We tried to take into account all the comments of the Reviewers and made appropriate changes to the text as well as added some Figures to the Supplementary materials. We also tried to correct the English of our article. Corrections were made to the word file in edit mode, and the changes are marked in color in the pdf file.

Now, we are sending the files of the updated text of the article (word and pdf) and Supplementary materials (pdf), the Cover letter file with our responses to the comments of the Reviewers and Figures 1 and 2 in new versions, to the Editorials.

The references in the text of the manuscript have been corrected: references 3, 4, 39, 63 and 69 have been added, some references to the authors' works have been removed (old ref. numbers: 42, 50, 53, 57, 65, 66). At the moment, self-citation does not exceed 15% of the total number of references (10 relative to 69).

Below we present the Authors' responses to the Reviewers' comments.

Reviewer 1.

  1. Page 5, they should also provide the HPLC characterisation of their peptides in the supporting information.

Response: Chromatograms are presented in the new version of the Supplementary materials (Figures S10.1–10.6).

  1. In the introduction, a couple of landmark summaries of Proline Rich AMPs are missing, such as CMLS, Cell. Mol. Life Sci.59, 1138–1150 (2002). https://doi.org/10.1007/s00018-002-8493-8, Amino Acids46, 2287–2294 (2014). https://doi.org/10.1007/s00726-014-1820-1

Response: We thank the Reviewer for this indication and regret that these reviews were missing by us. The references are included in the text.

Reviewer 2 Report

Comments and Suggestions for Authors

In this paper Tereshchenkov et al. reports the synthesis and antibacterial properties of TPP-AMP conjugates (bactenencin and oncocin derivatives). TPP is a well described ligand for targeting mitochondria as well as bacteria, and has previously been exploited for synthesis antibacterial conjugates. The paper suffers from poor readability and logical coherence of the presentation. Indeed, it is not clear what the scientific overall conclusion and significance of the work is. Basically the paper is close to mere data presentation as also seen in the title and in the abstract; and the conclusion section just repeats this. What is the aim and scientific progress and novelty that merit publication?

Some specific points

1)Figure 5 is slightly confusing, probably due to lack of technical information concerning the experiment/instrument. The technical part should be expanded.

29Table 2: cellular toxicity should report cell viability not growth inhibition.

3)Hemolysis is a standard parameter in antibiotic drug discovery, and should be included.

4)It is found that sbmA is responsible for the bacterial uptake of some of the compounds. This is a critical information in terms of possible resistance since sbmA is a nonessential protein, and thus easy resistance development would be predicted.

5)HPLC analysis data for all new compounds must be included I supplementary.

6)The in vitro translation data (Fig. 2B) should include dose response for proper comparison.

7)P16 , line 613: as I read the data of Fig. 5, membrane decrease is induced by comp. 2, although not very potently.

Comments on the Quality of English Language

The paper must be edited by a native English speaking scientist in the field

Author Response

We are grateful to the Reviewers for the careful reading of our manuscript, as well as for the valuable comments made. We tried to take into account all the comments of the Reviewers and made appropriate changes to the text as well as added some Figures to the Supplementary materials. We also tried to correct the English of our article. Corrections were made to the word file in edit mode, and the changes are marked in color in the pdf file.

Now, we are sending the files of the updated text of the article (word and pdf) and Supplementary materials (pdf), the Cover letter file with our responses to the comments of the Reviewers and Figures 1 and 2 in new versions, to the Editorials.

The references in the text of the manuscript have been corrected: references 3, 4, 39, 63 and 69 have been added, some references to the authors' works have been removed (old ref. numbers: 42, 50, 53, 57, 65, 66). At the moment, self-citation does not exceed 15% of the total number of references (10 relative to 69).

Below we present the Authors' responses to the Reviewers' comments.

Reviewer 2

The paper suffers from poor readability and logical coherence of the presentation. Indeed, it is not clear what the scientific overall conclusion and significance of the work is. Basically the paper is close to mere data presentation as also seen in the title and in the abstract; and the conclusion section just repeats this. What is the aim and scientific progress and novelty that merit publication?

Response: We thank the Reviewer for the critical view of our work. We have made some changes to the text of the manuscript, tried to place semantic accents and hope that the aim of the study and its novelty have become clearer.

Some specific points

1) Figure 5 is slightly confusing, probably due to lack of technical information concerning the experiment/instrument. The technical part should be expanded.

Response: The caption to the Figure 5 is changed:

“Figure 5. Effect of triphenylphosphonium analogs Bac(1-10, R/Y)-Cn-TPP (3: n = 2, 4: n = 10), TPP-C10-Bac(1-10, R/Y) (5), and the parent peptides 1 and 2 on the kinetics of the membrane potential of B. subtilis cells measured using the fluorescent probe diS-C3-(5). The dye is evenly distributed in the medium and fluorescence is observed. When cells are added, the dye accumulates inside living cells with potential, and the dye self-quenches and a decrease in fluorescence is observed. When substances that cause a decrease in membrane potential are added to cells, the dye is released into the medium, which causes an increase in fluorescence. Arrows mark moments at which appropriate amounts of compounds were added. The Gramicidin A at a concentration of 0.5 ng/mL was used as a control.”

2) Table 2: cellular toxicity should report cell viability not growth inhibition.

Authors response: Thank you for the note, there were measured the concentrations of the substances that cause 50% viability of cells.

The table caption was corrected.

3) Hemolysis is a standard parameter in antibiotic drug discovery, and should be included.

Response: Determining the effect of a substance on hemolysis is undoubtedly important for its use in medical practice. It is known that the introduction of alkyl(triphenyl)phosphonium (alkyl-TPP) moiety, for example, into liposomes, can affect hemolysis. If the TPP groups with a short alkyl chain do not affect hemolysis, than those with relatively long ones do (https://doi.org/10.1016/j.ijpharm.2021.120776; DOI: 10.1039/c9tb01853k). On the other hand, for ciprofloxacin derivatives containing hexyl-TPP, a slight effect on hemolysis has been shown compared with the parent antibiotic (doi: 10.3390/antibiotics9110758). So of course we will conduct hemolysis assay for our compounds in the future, perhaps not for those described in this manuscript, but for future ones based on these compounds.

4) It is found that sbmA is responsible for the bacterial uptake of some of the compounds. This is a critical information in terms of possible resistance since sbmA is a nonessential protein, and thus easy resistance development would be predicted.

Response:

That's right. However, our conjugates penetrate into bacterial cells not only through SbmA, but also due to the presence of alkyl-TPP, which easily penetrates through membranes. Therefore, if resistance occurs (due to mutations in the SbmA protein), this will be partially compensated by an alternative pathway into cells. This has been shown in our study. In addition, the participation of other membrane proteins in the transport of synthesized compounds cannot be excluded.

Despite the fact that the SbmA is not very specific to molecules that enter the bacterial cell through it, the fact that the conjugate of a short (deca) peptide with TPP penetrates into the bacterial cell via SbmA (as full-sized antimicrobial peptides) seems interesting to us in itself.

5) HPLC analysis data for all new compounds must be included in supplementary.

Response: Chromatograms are presented in the new version of the Supplementary materials (Figures S10.1–10.6).

6) The in vitro translation data (Fig. 2B) should include dose response for proper comparison.

Response: The dose-dependent curves of luciferase synthesis in a cell-free system in the presence of the studied compounds, as well as the corresponding diagram for several concentrations of compounds, are given in the new version of the Supplementary materials (Figure S3D). The reference to the Figure is given in the main text.

We have presented the results obtained in in vitro translation experiments for a concentration of 30 mkM, since in this concentration the difference between the degree of inhibition of translation in prokaryotic and eukaryotic systems is most evident. Besides, for in vitro translation experiments, a concentration of substances of 30 mkM was used, since it is 10 times or more higher than the concentration of ribosomes in the in vitro translation system, therefore potent translation inhibitors usually demonstrate almost 100% translation suppression in vitro at a concentration of 30 mkM. So, 30 mkM is a convenient concentration for evaluating the activity of compounds in vitro.

7) P16 , line 613: as I read the data of Fig. 5, membrane decrease is induced by comp. 2, although not very potently.

Response: You are absolutely right, peptide 2 slightly reduces the membrane potential. Unfortunately, the method does not allow us to unambiguously state by what mechanism the potential decrease occurs. This may be a protonophore effect, a protonophore-like effect, the formation of a physical pore (as in the case of gramicidin A), a change in the membrane conductivity for protons, blocking the functioning of components of the respiratory chain, or several other alternatives. To avoid speculation, we do not discuss this in this article.

Reviewer 3 Report

Comments and Suggestions for Authors

This article presents some data for synthesis and biological activities of newly synthesized Triphenylphosphonium analogs of short peptide related to bactenecin 7 and oncocin 112. The work has good scientific impact. However, I have some remarks to the authors:

1.      At many places authors make an abbreviation and they do not use further. Then why they do this abbreviation? For example, in ABSTRACT section at the beginning TPP is abbreviated and further the full name is used again, and many other places. Moreover, the abbreviations are one time given before full name and other time after. Please chose one of both and unify. For other abbreviations full names missed. Maybe they are clear for authors but it is not sure for the readers.

2.      What author means with 2-CTC resin, when they use this abbreviation and write 2-chlorotrityl resin? Further, author write Cl- loading of the resin. Do authors know that 2-CTC resin could exist in OH-form too, and if they mean chlorinated one, they have to use 2-chlorotrityl chloride resin. It is obviously from the experiments, but it is not clear in the manuscript text.

3.      In Figure 1A authors use underlining perhaps to show some moiety from the peptide. But generally underling is used in peptide chemistry for disulfide bridges which is not possible here, so please use normal font (NOT Bold) for peptide sequence and bold the needed sequence but do not underline.

4.      The columns for HPLC are described differently one time with ODS and the other with C18. It has the same meaning and so please unify and use one of both.

5.      Fig. S1 is absolutely not clear. Author use P at C-terminus may be for polymer, but often P is used for phosphate. And when they talk about the phosphate in the paper it is really not clear is this modified with phosphate peptide chain or it is a chain bonded to the polymer. Moreover, the figure is really not comprehensible. In line 194 authors write dibromoalkane but it is defined one dibromodecane. At many places authors direct to the other articles for some procedure but my opinion is that they have to be briefly presented. Readers could not to jump from article to article to search synthetic procedures.

6.      In line 162 authors wrote “The peptide was eluted in an ….”. Which peptide, authors present many new peptides here

7.      In the main text authors write litter and derivatives with L and in the supplementary files with l. Please unify.

8.      The MS data between lines 205 and 231 is better to be presented in table. They are not clear in a current presentation and make manuscript hard to be written in this place.

9.  It is very interesting that authors make the acetylation procedure with acetic anhydride 1 x 5 minutes and second time for 30 minutes. The opposite will be understandable but if it is the right way, why they make it at the beginning for such a short time in order to be obliged to do it secondly?

Finally, my opinion is that this article has very good scientific impact and it could be accepted for publication after correction of all remarks mentioned above.

Comments on the Quality of English Language

English language of the paper has to be revised at some places:

- There are many places where authors used active voice so they have to change to the passive voice (see lines 352, 364, etc.)

- Authors use very long sentences at some places and they are absolutely incomprehensible (see for example lines 362-367). For example, in line 105 The appearance of bacteria resistance to PA-AMP is not often observed, however the resistance can occur in the case of mutations or deletion of transporter protein genes required for the entry of PA-AMP into bacterial cells, in particular, sbmA [5,22,23,27] can be separated before However. Moreover, the manuscript is full with sentences where in one sentence many sub-sentences are included separated by ; Why they are not separated sentences? (see lines 91; 102; 127 and many others)

- What authors mean with by the way in line 103? It is not the needed sense so replace by with

Author Response

We are grateful to the Reviewers for the careful reading of our manuscript, as well as for the valuable comments made. We tried to take into account all the comments of the Reviewers and made appropriate changes to the text as well as added some Figures to the Supplementary materials. We also tried to correct the English of our article. Corrections were made to the word file in edit mode, and the changes are marked in color in the pdf file.

Now, we are sending the files of the updated text of the article (word and pdf) and Supplementary materials (pdf), the Cover letter file with our responses to the comments of the Reviewers and Figures 1 and 2 in new versions, to the Editorials.

The references in the text of the manuscript have been corrected: references 3, 4, 39, 63 and 69 have been added, some references to the authors' works have been removed (old ref. numbers: 42, 50, 53, 57, 65, 66). At the moment, self-citation does not exceed 15% of the total number of references (10 relative to 69).

Below we present the Authors' responses to the Reviewers' comments.

 Reviewer 3  

  1. At many places authors make an abbreviation and they do not use further. Then why they do this abbreviation? For example, in ABSTRACT section at the beginning TPP is abbreviated and further the full name is used again, and many other places. Moreover, the abbreviations are one time given before full name and other time after. Please chose one of both and unify. For other abbreviations full names missed. Maybe they are clear for authors but it is not sure for the readers.

Response: Corrected.

  1. What author means with 2-CTC resin, when they use this abbreviation and write 2-chlorotrityl resin? Further, author write Cl-loading of the resin. Do authors know that 2-CTC resin could exist in OH-form too, and if they mean chlorinated one, they have to use 2-chlorotrityl chloride resin. It is obviously from the experiments, but it is not clear in the manuscript text.

Response: Corrected to “2-chlorotrityl chloride resin”.

  1. In Figure 1A authors use underlining perhaps to show some moiety from the peptide. But generally underling is used in peptide chemistry for disulfide bridges which is not possible here, so please use normal font (NOT Bold) for peptide sequence and bold the needed sequence but do not underline.

Response: Figure 1A is corrected.

  1. The columns for HPLC are described differently one time with ODS and the other with C18. It has the same meaning and so please unify and use one of both.

Response: Corrected.

  1. Fig. S1 is absolutely not clear. Author use P at C-terminus may be for polymer, but often P is used for phosphate. And when they talk about the phosphate in the paper it is really not clear is this modified with phosphate peptide chain or it is a chain bonded to the polymer. Moreover, the figure is really not comprehensible.

Response: Figure S1 and figure caption are corrected. Added explanation of the symbol «P» in the circle.

In line 194 authors write dibromoalkane but it is defined one dibromodecane. At many places authors direct to the other articles for some procedure but my opinion is that they have to be briefly presented. Readers could not to jump from article to article to search synthetic procedures.

Response: We thank the Reviewer for drawing attention to the inaccuracy.

“NH2-Cn-TPP (n = 2, 10) were obtained in two stages from TPP by the conjugation of TPP with dibromoalkanes….” We are talking about two dibromoalkanes: dibromodecane and dibromoethane. Detailed synthetic procedures are presented in the Supplementary Materials. Corrections have been made to the text.

  1. In line 162 authors wrote “The peptide was eluted in an ….”. Which peptide, authors present many new peptides here.

Response: Corrected: “Onc112 was eluted in an aqueous gradient of CH3CN (from 5 to 55% for 17 min) with 0.1% TFA at a flow rate of 70 mL/min.”

  1. In the main text authors write litter and derivatives with L and in the supplementary files with l. Please unify.

Response: Corrected.

  1. The MS data between lines 205 and 231 is better to be presented in table. They are not clear in a current presentation and make manuscript hard to be written in this place.

Response: We thank the Reviewer for the suggestion. Detailed information on high-resolution mass spectra is presented in the form of a table and transferred to Supplementary materials (Table S1).

  1. It is very interesting that authors make the acetylation procedure with acetic anhydride 1 x 5 minutes and second time for 30 minutes. The opposite will be understandable but if it is the right way, why they make it at the beginning for such a short time in order to be obliged to do it secondly?

Response: The capping procedure is standard in the process of SPPS. In the manual version of the synthesis, this procedure is carried out in two steps: the first (short) replaces the washing with a capping solution, and the second is the real capping step.

- There are many places where authors used active voice so they have to change to the passive voice (see lines 352, 364, etc.)

Response: Corrected. We changed the active voice to the passive voice in most of the sentences where it met.

- Authors use very long sentences at some places and they are absolutely incomprehensible (see for example lines 362-367). For example, in line 105 The appearance of bacteria resistance to PA-AMP is not often observed, however the resistance can occur in the case of mutations or deletion of transporter protein genes required for the entry of PA-AMP into bacterial cells, in particular, sbmA [5,22,23,27] can be separated before However. Moreover, the manuscript is full with sentences where in one sentence many sub-sentences are included separated by ; Why they are not separated sentences? (see lines 91; 102; 127 and many others).

Response: Corrected.

- What authors mean with by the way in line 103? It is not the needed sense so replace by with

Response: Corrected.

 R

Reviewer 4 Report

Comments and Suggestions for Authors

It is an interesting article about Triphenylphosphonium analogs of short peptide related to 2 bactenecin 7 and oncocin 112 as antimicrobial agents. The article is generally well written and easy to follow, the results are well presented.

However, the analysis of toxicity results (MTT) should be deeper and more critical- the GI50 values should be compared with MIC values. Also, the value of GI50 in cells does not imply the absence of toxic effect, the absence of toxic effects could be assumed at 80-90% viable cells not at 50%.

Line 647: please indicate the concentrations at which toxic effect was absent

Author Response

Reviewer 4

We apologize to Reviewer 4, we received your review after we corrected the text according to the comments of Reviewers 1, 2, 3 and 5. Therefore, corrections based on your comments were made in a later file (see, please, files 2024.01.06_corrections_color_pharmaceutics-2730252.docx and 2024.01.06_color_pharmaceutics-2730252.pdf) and along with corrections of the similarity of the text with previous articles.

Comments:

However, the analysis of toxicity results (MTT) should be deeper and more critical- the GI50 values should be compared with MIC values. Also, the value of GI50 in cells does not imply the absence of toxic effect, the absence of toxic effects could be assumed at 80-90% viable cells not at 50%.

Response:

Thank you for the note. To show the absence of cytotoxic effects at the MIC concentrations we added concentration-viability dependencies after treatment by tested compounds to the Supplementary materials (Section V). We would prefer to leave the IC50 values in the manuscript, since they are more familiar to a wide range of readers.

Line 647: please indicate the concentrations at which toxic effect was absent

Response: Of course, the most of the compounds became toxic at high concentrations, thus we add the note to the text that the compounds were not toxic for human cell lines at MIC concentrations:

“The absence of a toxic effect (Table 2, Figure S11) at MIC concentrations in combination with an inhibitory effect on the bacterial translation process……….”

Reviewer 5 Report

Comments and Suggestions for Authors

Authors present study entitled Triphenylphosphonium analogs of short peptide related to bactenecin 7 and oncocin 112 as antimicrobial agents, and as a result suggest that the structural features of triphenylphosphonium analogs of short peptide related to Bac7 and Onc112 can be applied for the creation of therapeutic antibacterial agents based on AMP.

I only strongly suggest to adress in more details the stability of a newly synthetised compound and the activity against microorganisms.

Minor comment: There is extensive citation of a previous papers published by the first author which I think can be avoided.

Author Response

Reviewer 4

I only strongly suggest to address in more details the stability of a newly synthesised compound and the activity against microorganisms.

Response: We have tried to discuss in more detail the activity of new compounds on bacterial strains.

According to our preliminary data, the incorporation of TPP in peptide structure improves the stability of the decapeptide in the mice serum. However, the stability of compounds is a separate study that we are currently conducting. This study includes determining the stability of the obtained compounds, as well as the synthesis of new analogues that should be more stable in biological media. We are going to publish results of this study separately.

Minor comment: There is extensive citation of a previous papers published by the first author which I think can be avoided.

Response: Corrected (please see the preamble to the responses to the comments of all Reviewers).

Round 2

Reviewer 2 Report

Comments and Suggestions for Authors

The revised version of the manuscript is significantly improved, and most of the technical queries have been addressed. However, it is still mainly a data presentation paper without major conclusions or visions of the significance of the results, and as a drug discovery study it is far from comprehensive. The improvements in term of additional experiments to strengthen the paper suggested by the referees have not been implemented, but additional clarifying data (e.g. relating to translation inhibition) have been included  (primarily in Supplementary). Although the paper has also been improved in terms of language and readability, the manuscript still requires language improvement, and should be edited by a native English speaking scientist within the field.

As a technical suggestion, the authors should consider using the more direct Sytox Deep Red assay for membrane disruption determination instead of Dis-C3-(10) used.

Comments on the Quality of English Language

The manuscript still requires language improvement, and should be edited by a native English speaking scientist within the field.

Reviewer 4 Report

Comments and Suggestions for Authors

The authors have satisfactorily answered my comments. I recommend publication of this article in Pharmaceutics.